# Triterpenoids from the Leaves of *Cyclocarya paliurus* and Their Glucose Uptake Activity in 3T3-L1 Adipocytes

**DOI:** 10.3390/molecules28083294

**Published:** 2023-04-07

**Authors:** Xiaoqin Liang, Shengping Deng, Yan Huang, Liwei Pan, Yanling Chang, Ping Hou, Chenyang Ren, Weifeng Xu, Ruiyun Yang, Kanyuan Li, Jun Li, Ruijie He

**Affiliations:** 1State Key Laboratory for Chemistry and Molecular Engineering of Medicinal Resources/Key Laboratory for Chemistry and Molecular Engineering of Medicinal Resources (Ministry of Education of China), Collaborative Innovation Center for Guangxi Ethnic Medicine, School of Chemistry and Pharmaceutical Sciences, Guangxi Normal University, Guilin 541004, Chinayang_rui_yun@163.com (R.Y.);; 2Guangxi Key Laboratory of Plant Functional Phytochemicals and Sustainable Utilization, Guangxi Institute of Botany, Guangxi Zhuang Autonomous Region and Chinese Academy of Sciences, Guilin 541006, China

**Keywords:** *Cyclocarya paliurus*, dammarane triterpenoid saponins, glucose uptake, 3T3-L1 adipocytes

## Abstract

Four new dammarane triterpenoid saponins cypaliurusides **Z_1_**–**Z_4_** (**1**–**4**) and eight known analogs (**5**–**12**) were isolated from the leaves of *Cyclocarya paliurus*. The structures of the isolated compounds were determined using a comprehensive analysis of 1D and 2D NMR and HRESIMS data. The docking study demonstrated that compound 10 strongly bonded with PTP1B (a potential drug target for the treatment of type-II diabetes and obesity), hydrogen bonds, and hydrophobic interactions, verifying the importance of sugar unit. The effects of the isolates on insulin-stimulated glucose uptake in 3T3-L1 adipocytes were evaluated and three dammarane triterpenoid saponins (**6**, **7** and **10**) were found to enhance insulin-stimulated glucose uptake in 3T3-L1 adipocytes. Furthermore, compounds **6**, **7,** and **10** exhibited potent abilities to promote insulin-stimulated glucose uptake in 3T3-L1 adipocytes in a dose-dependent manner. Thus, the abundant dammarane triterpenoid saponins from *C. paliurus* leaves exhibited stimulatory effects on glucose uptake with application potential as a antidiabetic treatment.

## 1. Introduction

Triterpenoids, with a wide variety of structural types and extensive biological activity, are important compounds for the prevention and treatment of various kinds of diseases, such as diabetes mellitus and metabolic syndrome, cancer, hyperlipidemia, cardiovascular and cerebrovascular disease, neurodegenerative disease, bone disease, liver disease, kidney disease, aging, gastrointestinal disease, mental illness involving depression, and skin aging [1,2,3,4,5]. Among triterpenoids, dammarane triterpenoids exhibit the most significant hypoglycemic activity [6,7,8].

*Cyclocarya paliurus*, an endemic plant that grows in southern China, has been widely used as an herbal tea and is commonly known as the ‘sweet tea tree’. The leaves of this plant have been used in Chinese folk medicine to prevent diabetes, hypertension, inflammation, and heart disease [9,10]. There have been many reports on the significant antidiabetic effects of the extract of *C. paliurus* leaves [11,12,13,14,15,16,17]. An antidiabetic prescription containing the leaves of *C. paliurus* reduces blood glucose and improves glucose tolerance [18,19]. The leaves of *C. paliurus* were approved as a novel raw food by the National Health Commission of the People’s Republic of China in 2013 [20]. Currently, *C. paliurus* is a crop for rural revitalization and an important medical and functional raw food. A variety of triterpenoids, flavonoids, and other compounds can be isolated from the leaves of this plant [9,10]. Among these constituents, dammarane triterpenoids are characteristic indicators and the key functional and active constituents of *C. paliurus* [9,10,21,22]. Several dammarane triterpenoids from this plant demonstrated to moderate the activity of PTP1B (a potential drug target for the treatment of type-II diabetes and obesity) and enhance insulin-stimulated glucose uptake in 3T3-L1 adipocytes [23,24,25,26]. To further investigate potential constituents for promoting glucose uptake activity in 3T3-L1 adipocytes, the active ingredients were separated from the leaves of *C. paliurus* to afford four novel dammarane saponins (**1**−**4**) and eight known analogs (**5**−**12**) (Figure 1). Furthermore, stimulation of glucose consumption in 3T3-L1 adipocytes by the triterpenoids isolated from *C. paliurus* leaves was evaluated and possible hypoglycemic mechanisms of the active compounds were studied. Herein, the isolation, purification, and determination of these isolates, as well as the assays used to determine the glucose uptake activity in 3T3-L1 adipocytes of the constituents and predictions of their mechanisms, are described.

## 2. Results and Discussion

### 2.1. Elucidation of the Chemical Structure of Cypaliurusides Z_1_–Z_4_ (***1**−**4***)

Cypaliuruside Z_1_ (**1**) was deduced to have the molecular formula C_37_H_58_O_12_ (Appendix A) based on its NMR (^1^H, ^13^C, pyridine-*d*_5_) data and the positive-ion peak [M+Na]^+^ at *m*/*z* 717.3811 (calculated for C_37_H_58_O_12_Na, 717.3826) in its HRESIMS spectrum. Comparisons between the MS and NMR signals of compound **1** and those of cyclocarioside L (**5**) [27] indicated that **1** and **5** have almost identical NMR data (Table 1; Appendix A). There were only a few differences in the ^1^H-and ^13^C-NMR data (pyridine-*d*_5_) between **1** and **5** at the positions 20, 22, 23, and 24 and two sugar units. The ^1^H NMR signals of H-22 and 23 shifted from *δ*_H_ 1.97 (m, H-22), 1.82 (m, H-22), 2.64 (m, H-23), and 2.49 (m, H-23) for **5** to *δ*_H_ 7.64 (d, 5.7, H-22) and 6.26 (d, 5.7, H-23) for **1**, respectively (Table 1; Appendix A). The ^13^C NMR (pyridine-*d*_5_, Table 1; Appendix A) signals of C-20, C-22, C-23, and C-24 shifted from *δ*_C_ 89.6, 32.5, 29.7, and 176.9 for **5** to *δ*_C_ 92.6, 161.2, 121.9, and 173.4 for **1**, respectively (Table 1); this indicates that there is a double bond at the positions 22 and 23 in **1**. Moreover, the ^13^C NMR data of the two sugar units in **1** contained ten carbon signals is compared to eleven corresponding carbon signals for **5**. In the ^1^H-^1^H COSY spectrum of **1**, the correlations of H-1/H-2/H-3, H-5/H-6/H-7 and H-9/H-11/H-12/H-13/H-15/H-16/H-17 substantiated the existence of three fragments: –CH_2_CH_2_CH–, –CH_2_CH_2_CH–, and –CHCH_2_CHCHCHCH CH_2_– (Figure 2A and Appendix A). In the HMBC spectrum of **1** (Figure 2A and Appendix A, pyridine-*d_5_*), the cross peaks from H-19-CH_3_ (*δ*_H_ 1.29, s) to C-1 (*δ*_C_ 36.1), C-5 (*δ*_C_ 51.3) and C-9 (*δ*_C_ 54.3), H-18-CH_3_ (*δ*_H_ 0.90, s) to C-9 (*δ*_C_ 54.3) and C-14 (*δ*_C_ 41.9), and H-30-CH_3_ (*δ*_H_ 0.55, s) to C-8 (*δ*_C_ 50.7) and C-15 (*δ*_C_ 31.7) indicated that CH_3_-19 was linked to C-10, CH_3_-18 was located at C-8, and CH_3_-30 was connected to C-14, respectively. The HMBC correlations from H-21 (*δ*_H_ 1.36, s) to C-17 (*δ*_C_ 47.8) and C-22 (*δ*_C_ 121.9) suggested that CH_3_-21 was situated at C-20. Furthermore, the diagnostic ROESY correlations of H-3 and H-11 to H-19 suggested that H-3, H-11, and H-19 are *β*-oriented, biosynthetic characteristics of dammarane trierpenoids [28], further indicating that H-19 just located at *β*-oriented. Similarly, the diagnostic ROESY peaks of H-17 and H-30 indicated that H-17 and H-30 are *α*-oriented (Figure 2B). The HMBC correlations from the anomeric protons H-1′ (*δ*_H_ 4.73, d, *J* = 6.8 Hz) to C-3 (*δ*_C_ 82.2) and H-1″ (*δ*_H_ 4.87, d, *J* = 6.5 Hz) to C-12 (*δ*_C_ 76.8) suggest that the two L-arabinoses are situated at C-3 and C-12 respectively (Figure 2A). The sugar unit was determined to be an *α*-L-arabinosyl moiety based on the hydrolysis of **1** with 1 M HCl and HPLC (Table 2). Therefore, **1** was deduced to have the structure (20*S*,24)-lactone-22-en-dammarane-(3*α*,12*β*)-12-*O*-*α*-L-arabinopyranosyl-3-*O*-*α*-L-arabinopyranoside and, therefore, is named cypaliuruside Z_1_ (Figure 1).

Cypaliuruside Z_2_ (**2**) was deduced to have the molecular formula C_38_H_62_O_13_ by HRESIMS *m*/*z* 749.4075 [M + Na]^+^ (calculated for C_38_H_62_O_13_Na, 749.4088) (Appendix A), suggesting eight degrees of unsaturation. Detailed analyses show that the NMR data of **2** (Table 1) were highly similar to those of **5** [27], suggesting that **2** includes a basic (20*S*,24)-lactone-22-en-dammarane triterpenoid skeleton and differed from those of cyclocarioside L only in the signals of the two sugar units (Table 1; Appendix A). The ^1^H NMR spectrum indicated two anomeric protons at *δ*_H_ 5.11 (d, *J* = 7.7 Hz, H-1″) and 4.74 (d, *J* = 6.5 Hz, H-1′) and six methyl protons at *δ*_H_ 1.38 (s, H-19), 1.26 (s, H-29), 0.98 (s, H-18), 0.96 (s, H-28), 1.28 (s, H-21), and 0.62 (s, H-30) (Table 1). The ^13^C NMR of 38 carbon resonances confirmed the aforementioned moieties (Table 1). The ^13^C-NMR (pyridine-*d*_5_, Table 1, Appendix A) and DEPT (distortionless enhancement by polarization transfer, pyridine-*d*_5_) spectra indicated the presence of six methyl carbon signals at *δ*_C_ 17.5 (C-18), 17.4 (C-19), 24.9 (C-21), 23.8 (C-28), 30.0 (C-29), and 16.8 (C-30) and two glycosyl anomeric carbon signals at *δ*_C_ 101.8 (C-1′) and *δ*_C_ 102.3 (C-1″), respectively. In the ^1^H-^1^H COSY spectrum of **2**, the correlations of H-1/H-2/H-3, H-5/H-6/H-7 and H-9/H-11/H-12/H-13/H-15/H-16/H-17 substantiated the existence of three fragments: –CH_2_CH_2_CH–, –CH_2_CH_2_CH–, and –CHCH_2_CHCHCHCH_2_CH_2_– (Figure 2A and Appendix A). The sugar unit was determined to be *α*-L-arabinopyranose and *β*-D-glucopyranose based on the hydrolysis of **2** with 1 M HCl and HPLC (Table 2). Therefore, **2** was deduced to have the structure (20*S*,24)-lactone-dammarane-(3*α*,12*β*)-12- *O*-*β*-D-glucopyranosyl-3-*O*-*α*-L-arabinopyranoside and is, thus, named cypaliuruside Z_2_ (Figure 1).

The molecular formula of cypaliuruside Z_3_ (**3**) was deduced to be C_38_H_62_O_13_ due to the positive sodium adduct ion HRESIMS peak at *m*/*z* 749.4096 [M + Na]^+^ (calculated for C_38_H_62_O_13_Na, 749.4088) (Appendix A), which is the same as that of **2**. The ^1^H- and ^13^C-NMR data of **3** (pyridine-*d*_5_, Table 3) were found to be identical to those of **2**. A small difference between the ^1^H- and ^13^C-NMR data of **3** and **2** (pyridine-*d*_5_) was related to a sugar unit linked to C-3. That is, the ^1^H-NMR data of the sugar unit linked to C-3 shifted from *δ*_H_ 4.74 (d, *J* = 6.5 Hz, H-1′), 4.45 (t, *J* = 8.9 Hz, H-2′), 4.27 (dd, *J* = 8.4, 3.5 Hz, H-3′), 4.42 (m, H-4′), 4.34 (dd, *J* = 12.1, 3.7 Hz, H-5′), and 3.80 (m, H-5′) for **2** (pyridine-*d*_5_, Table 2) to *δ*_H_ 5.55 (d, *J* = 6.5 Hz, H-1′), 4.89 (t, *J* = 5.6 Hz, H-2′), 4.85 (m, H-3′), 4.76 (dd, *J* = 6.0, 5.0 Hz, H-4′), 4.40 (m, H-5′), and 4.29 (d, *J* = 9.4 Hz, H-5′) for **3** (pyridine-*d*_5_, Table 2). Moreover, the ^13^C-NMR data of the sugar unit linked to C-3 changed from *δ*_C_ 102.3 (C-1′), 72.1 (C-2′), 75.3 (C-3′), 69.7 (C-4′), and 66.9 (C-5′) for **2** to *δ*_C_ 106.9 (C-1′), 84.6 (C-2′), 79.9 (C-3′), 86.0 (C-4′), and 63.4 (C-5′) for **3**, respectively (pyridine-*d_5_*, Table 2). Comparing the ^1^H- and ^13^C-NMR (Table 2) and HRESIMS data between **3** and **2** revealed the presence of an arabinofuranose linked to C-3 in **3** instead of an arabinopyranose linked to C-3 in **2**. *α*-L-Arabinofuranose and *β*-D-glucopyranose were confirmed by the acid hydrolysis solution of **3** in conjunction with comparison to authentic sugars in the HPLC assay (Table 2). Accordingly, **3** was deduced to have the structure (20*S*,24)-lactone-dammarane-(3*α*,12*β*)-12-*O*-*β*-D-glucopyranosyl-3-*O*-*α*-L-arabino-furanoside and is, thus, named cypaliuruside Z_3_ (**3**) (Figure 1).

Cypaliuruside Z_4_ (**4**) was deduced to have the molecular formula C_38_H_62_O_12_ based on the positive sodium adduct ion HRESIMS peak at *m*/*z* 733.4146 [M + Na]^+^ (calculated for C_38_H_62_O_12_Na, 733.4139) (Appendix A), which is 16 Da lower than that of **2**. A detailed analysis of the ^1^H- and ^13^C-NMR data (pyridine-*d*_5_, Table 3) showed that **4** is very similar to **2**, except that the glucopyranosyl moiety is replaced by a quinovopyranosyl moiety at C-12 in **4**. Moreover, the hydrolysate of compound **4** was determined to contain L-arabinose and quinovose by comparison with an authentic sample in the HPLC test (Table 2). Accordingly, **4** was deduced to have the structure (20*S*,24)-lactone-dammarane-(3*α*,12*β*)-12-*O*-*β*-D-quinovopyranosyl-3-*O*-*α*-L-arabinopyranoside and is, thus, named cypaliuruside Z_4_ (Figure 1).

By comparing the measured NMR (^1^H and ^13^C) and MS data to those reported in the literature, the known dammarane triterpenoid saponins were identified as cyclocarioside L (**5**) [27], cypaliuruside M (**6**) [29], cyclocarioside J (**7**) [30], cyclocarioside Z_18_ (**8**) [25], cyclocarioside Z_10_ (**9**) [25], cyclocarioside H (**10**) [31], 2*α*,3*α*,23-trihydroxy-12,20(30)-dien-28-ursolic acid 28-*O*-*β*-D-glucopyranoside (**11**) [32], 1-oxo-3*β*,23-dihydroxy olean-12-en-28-oic acid 28-*O*-*β*-D-glucopyranoside (**12**) [33].

### 2.2. Predicted Binding Modes of Compounds and PTP1B Using Molecular Docking Analysis

PTP1B is a potential drug target for the treatment of type-II diabetes and obesity. To to investigate the interactions of PTP1B with dammarane triterpenoid saponins, an independent docking run was performed and the compounds ligand with the lowest binding affinity mode was selected for analysis. 

The results showed that **10** was well docked into the active site in PTP1B and the binding energy was −8.9 kcal·mol^−1^, which is superior to that of the oleanolic acid (positive control, −7.2 kcal·mol^−1^). The catalytic and allosteric sites of docking compound **10** with PTP1B were further simulated. The results showed that compound **10** depicted multiple important interactions, such as Pi-sigma, alkyl and Pi-alkyl, conventional hydrogen bonds, and van der Waals interactions within the active pocket of PTP1B (Figure 3). In visualization of the catalytic site docking results (Figure 3A,C), various amino acid residues, such as Asp548, Ser618, Lys616, Lys620, Asp681, Gly718, Ile719, Ala717, Asp715, Gly720, Ser716, Arg721, Gln762, Phe682, Tyr546, and Val549, surrounded the active pocket of PTP1B. Hydroxyls of compound **10** mainly formed hydrogen bonds with the PTP1B residues Gly720, Asp715, Ile719, ASP681, and Gln762. Moreover, Phe628 revealed Pi-sigma and Pi-alkyl interaction with the methyl of C-6′ and C-30, respectively. In visualization of the allosteric site docking results, the sugar unit of compound **10** formed hydrogen bonding with the PTP1B residue Asp789, whereas van der Waals interactions were noticed with Ser786 and Gln788. In addition, the hydrophobic part of the ligand revealed Alkyl and Pi-alkyl interaction with the Phe780 and Lys792 (Figure 3B,D).

### 2.3. EtOAc Extract and Dammarane Saponins Enhance Glucose Uptake

The 3T3-L1 preadipocyte is one of the most commonly used in vitro models for screening antidiabetic compounds [34,35]. In the present study, the enhancement of glucose uptake by the EtOAc extract (GAE) and triterpenoid saponins (**1**–**12**) of *C. paliurus* was investigated using a 2-(N-(7-nitrobenz-2-oxa-1,3-diazol-4-yl)amino)-2-deoxy glucose (2-NBDG) uptake model.

Firstly, the cell viability was measured using the MTT assay [3-(4,5-dimethylthiazol-2-yl)-2,5-diphenyltetrazolium bromide]. The 3T3-L1 adipocyte cell viability test showed the extract at a glucose: 2-NBDG. Isolated compounds and rosiglitazone (ROS, the positive control) were treated to differentiate concentrations of 25 μg/mL; the triterpenoid saponins (**1**–**12**) at a concentration of 10 μM were not cytotoxic and the cells had a survival rate of up to 90%.

Thus, the extract of *C. paliurus* leaves (25 μg/mL) and isolated triterpenoid saponins (**1**–**12**, 10 μM) were added to differentiated 3T3-L1 adipocytes with 2-NBDG. As shown in Figure 4A, all isolated triterpenoid saponins (**1**–**12**) exhibited potency to enhance glucose uptake in 3T3-L1 adipocytes. Among the isolates, compounds **6**–**7** and **10** at 10 μM increased insulin-stimulated glucose uptake by approximately 37%, 35%, and 46% in 3T3-L1 adipocytes, respectively. The positive control rosiglitazone (ROS, 10 μM) increased glucose uptake in the 3T3-L1 adipocytes by approximately 55%. These triterpene saponins enhance glucose uptake in 3T3-L1 adipocytes more effectively than those of the extract of *C. paliurus* leaves. The active **6**–**7** and **10** at various concentrations (2.5, 5, and 10 μM) were further investigated for their effect on glucose uptake. As shown in Figure 4B, **6**–**7** and **10** significantly increase glucose uptake in the 3T3-L1 adipocytes compared with the vehicle control, which indicates that these compounds may enhance glucose uptake by improving insulin sensitivity in a concentration-dependent manner.

## 3. Materials and Methods

### 3.1. General Experimental Procedures

Silica gel (200–300 mesh, Qingdao Marine Chemical Co., Ltd., Qingdao, China), reversed-phase C18 (50 μm, Merck, Rahway, NJ, USA), and Sephadex LH-20 (Amersham Pharmacia Biotech, Amersham, UK) columns were used for chromatographic separations. L-Arabinose, D-arabinose, L-glucose, D-glucose, L-quinovose and D-quinovose were used as sugar standards (Sigma, Munich, Germany). All solvents were of HPLC or analytical grade. Agilent 1260 semipreparative HPLC (ODS-A, 8 μm, 250 ×10.0 mm, YMC, Kyoto, Japan).

### 3.2. Plant Material

Dried leaves (10 kg) of *C. paliurus* were obtained from Xiushui County, Jiangxi, China, in July 2020 and identified by Associate Professor Qiang Xie (Guangxi Normal University). A voucher specimen (No. ID-202070910) was deposited at the State Key Laboratory of Guangxi Normal University (GXNU), China.

### 3.3. Computational Analysis of Molecular Docking Simulation

Molecular docking was used to investigate the interactions between triterpenoids and PTP1B. The structure of PTP1B (PDB ID: 1EEO [36]) was obtained from the Online Protein Data Bank and the structures of the ligands were downloaded from Pubchem. AutoDock tools (ADT, version: 1.5.6) to explore the binding mode between PTP1B and the ligands. AutoDock vina (version 1.2.3) calculated scoring function and predicted binding affinity (kcal/mol), while Pymol (version 2.2.0) was used to analyze the visualization of the docking results.

### 3.4. Extraction and Separation of Dammarane Saponins

The leaves of *C. paliurus* (10 kg) were purchased from Liuhe medicine market of Guilin city in October 2020. The leaves were extracted 3 times with a 75% ethanol–H_2_O mixture (3:1, *v*/*v*, 20 L) under reflux and concentrated under vacuum to remove the ethanol, obtaining a crude extract (2.5 kg). The crude extract (2.5 kg) was suspended in distilled water and partitioned with polyethylene (PE), ethyl acetate (EtOAc), and n-butanol (n-BuOH) to obtain a PE extract (480 g), EtOAc extract (450 g), and *n*-BuOH extract (1090 g). The glucose uptake activity of the sub-extracts in 3T3-L1 adipocytes was assessed.

The active *n*-BuOH fraction (1090 g) was chromatographed on a silica gel column eluted with a gradient of increasing methanol content (0–100%) in a mixture with dichloromethane to yield 8 fractions (Fr. I to Fr. VIII). The active Fr. III (220 g) was subjected to C_18_ column chromatography (H_2_O/MeOH 50:50–10:90) to provide nine fractions (Fr. III-1 to Fr. III-9). Fr. III-2 (2.3 g) was fractioned by an HW-40F column (H_2_O/MeOH, 100:0−0:100) to obtain four subfractions (Fr. III-2-1 to Fr. III-2-7). Fr. III-2-1 (1000 mg) was isolated with a C_18_ column, followed by C_18_ semipreparative HPLC (H_2_O-MeCN, 34:66, *v*/*v*, 3.0 mL/min), to yield **1** (t_R_ 28.43 min, 10.7 mg), **2** (t_R_ 18.59 min, 3.2 mg), and **3** (t_R_ 22.68 min, 5.1 mg). Fr. III-3 (4.0 g) was separated by silica gel (CH_2_Cl_2_-MeOH, 100:0–0:100) to obtain nine subfractions (Fr. III-3-1 to Fr. III-3-9). Fr. III-3-6 was purified by Sephadex LH-20 (MeOH) and C_18_ semipreparative HPLC (H_2_O-MeCN, 58:42, *v*/*v*, 3.0 mL/min) to obtain **4** (t_R_ 15.10 min, 3.5 mg), **11** (t_R_ 35.46 min, 6.8 mg) and **12** (t_R_ 45.12 min, 2.7 mg). Fr. III-6 (3.8 g) was further chromatographed by silica gel CC eluted with CH_2_Cl_2_-MeOH (12:1, 10:1, 6:1) to 100% to afford three subfractions (Fr. III-6-1 to Fr. III-6-3). Fr. III-6-2 was subjected to Sephadex LH-20 CC, eluted with MeOH, and then further purified by C_18_ preparative HPLC (H_2_O-MeOH, 15:85, *v*/*v*, 3.0 mL/min) to afford compounds **7** (t_R_ 25.35 min, 3.0 mg), **9** (t_R_ 13.05 min, 3.7 mg), and **10** (t_R_ 16.89 min, 2.0 mg). Fr. III-6-3 was isolated by Sephadex LH-20, eluted with MeOH, and then purified by C_18_ preparative HPLC (H_2_O-MeOH, 15:85, *v*/*v*, 3.0 mL/min) to obtain compounds **5** (t_R_ 16.89 min, 2.0 mg), **6** (t_R_ 16.78 min, 3.5 mg), and **8** (t_R_ 22.23 min, 10.3 mg).

### 3.5. Characterization of the Isolates

Cypaliuruside Z_1_ (**1**): white amorphous powder; ${\left[\alpha \right]}_{D}^{25}$= −18.72 (*c* 0.30, MeOH); HRESIMS *m*/*z* 717.3811 [M + Na]^+^ (calculated for C_37_H_58_O_12_Na, 717.3826); for the ^1^H (pyridine-*d*_5_, 500 MHz) and ^13^C NMR (pyridine-*d*_5_, 125 MHz) data, see Table 1. All significant data are given in the electronic Appendix A.

Cypaliuruside Z_2_ (**2**): white amorphous powder;
${\left[\alpha \right]}_{D}^{25}$= −20.03 (*c* 0.22, MeOH); HRESIMS *m*/*z* 749.4075 [M + Na]^+^ (calculated for C_38_H_62_O_13_Na, 749.4088); for the ^1^H (pyridine-*d*_5_, 600 MHz) and ^13^C NMR (pyridine-*d*_5_, 150 MHz) data, see Table 1. All significant data are given in the electronic Appendix A.

Cypaliuruside Z_3_ (**3**): white amorphous powder; ${\left[\alpha \right]}_{D}^{25}$= −20.94 (*c* 0.22, MeOH); HRESIMS *m*/*z* 749.4096 [M + Na]^+^ (calculated for C_38_H_62_O_13_Na, 749.4088); for the ^1^H (pyridine-*d*_5_, 500 MHz) and ^13^C NMR (pyridine-*d*_5_, 125 MHz) data, see Table 2. All significant data are given in the electronic Appendix A.

Cypaliuruside Z_4_ (**4**): white amorphous powder; ${\left[\alpha \right]}_{D}^{25}$= −19.89 (*c* 0.25, MeOH); HRESIMS *m*/*z* 733.4146 [M + Na]^+^ (calculated for C_38_H_62_O_12_Na, 733.4139); for the ^1^H (pyridine-*d*_5_, 600 MHz) and ^13^C NMR (pyridine-*d*_5_, 150 MHz) data, see Table 2. All significant data are given in the electronic Appendix A.

Cyclocarioside L (**5**): white amorphous powder. ^1^H NMR (pyridine-*d*_5_, 600 MHz) *δ*_H_ 2.98 (1H, m, H-1a), 2.01 (1H, m, H-1b), 2.01 (1H, m, H-2a), 1.89 (1H, m, H-2b), 3.50 (1H, m, H-3), 1.53 (1H, d, *J* = 11.0 Hz, H-5), 1.63 (1H, m, H-6a), 1.45 (1H, m, H-6b), 1.39 (1H, m, H-7a), 1.08 (1H, d, *J* = 9.2 Hz, H-7b), 4.16 (1H, m, H-12), 0.92 (3H, s, H-18), 1.35 (3H, s, H-19), 1.45 (3H, s, H-21), 0.93 (3H, s, H-28), 1.23 (3H, s, H-29), 0.65 (3H, s, H-30), 5.45 (1H, d, *J* = 7.0 Hz, H-1′), 4.75 (1H, dd, *J* = 3.1, 8.5 Hz, H-4′), 5.10 (1H, d, *J* = 7.5 Hz, H-1″), 1.65 (3H, d, *J* = 7.3 Hz, H-6″). ^13^C NMR (pyridine-*d*_5_, 150 MHz) *δ*_C_ 37.1 (C-1), 22.4 (C-2), 80.5 (C-3), 38.9 (C-4), 50.7 (C-5), 19.3 (C-6), 36.0 (C-7), 50.6 (C-8), 57.7 (C-9), 40.3 (C-10), 34.2 (C-11), 76.1 (C-12), 41.6 (C-13), 40.5 (C-14), 31.7 (C-15), 25.7 (C-16), 50.1 (C-17), 17.0 (C-18), 17.1 (C-19), 89.6 (C-20), 21.4 (C-21), 32.1 (C-22), 30.3 (C-23), 176.2 (C-24), 23.6 (C-28), 30.4 (C-29), 16.8 (C-30), 102.4 (C-1′), 72.3 (C-2′), 75.4 (C-3′), 69.4 (C-4′), 69.8 (C-5′), 101.5 (C-1″), 75.1 (C-2″), 78.7 (C-3″), 78.1 (C-4″), 72.1 (C-5″), 18.9 (C-6″).

Cypaliuruside M (**6**): white amorphous powder. ^1^H NMR (pyridine-*d*_5_, 600 MHz) *δ*_H_ 3.12 (1H, m, H-1a), 2.11 (1H, m, H-1b), 1.97 (1H, m, H-2a), 1.59 (1H, m, H-2b), 3.64 (1H, t, *J* = 2.7 Hz, H-3), 1.68 (1H, m, H-5), 1.53 (1H, m, H-6a), 1.45 (1H, m, H-6b), 1.49 (1H, m, H-7a), 1.28 (1H, m, H-7b), 4.26 (1H, dt, *J* = 4.7, 10.5 Hz, H-11), 1.12 (3H, s, H-18), 1.45 (3H, s, H-19), 1.15 (3H, s, H-21), 1.45 (3H, s, H-26), 1.43 (3H, s, H-27), 1.03 (3H, s, H-28), 1.33 (3H, s, H-29), 0.65 (3H, s, H-30), 4.65 (1H, d, *J* = 7.5 Hz, H-1′), 4.21 (1H, dd, *J* = 2.1, 8.5 Hz, H-3′), 4.25 (1H, dd, *J* = 5.1, 9.5 Hz, H-4′), 4.35 (1H, dd, *J* = 5.1, 11.5 Hz, H-5′), 1.59 (3H, d, *J* = 6.7 Hz, H-6′), 5.05 (1H, d, *J* = 7.7 Hz, H-1″), 4.05 (1H, d, *J* = 5.3 Hz, H-2″), 4.10 (1H, dd, *J* = 5.1, 11.5 Hz, H-3″), 1.68 (3H, d, *J* = 5.3 Hz, H-6″). ^13^C NMR (pyridine-*d*_5_, 150 MHz) *δ*_C_ 36.1 (C-1), 21.4 (C-2), 81.5 (C-3), 38.9 (C-4), 50.9 (C-5), 18.3 (C-6), 36.7 (C-7), 50.6 (C-8), 54.7 (C-9), 40.6 (C-10), 76.2 (C-11), 34.1 (C-12), 41.6 (C-13), 40.8 (C-14), 31.4 (C-15), 26.7 (C-16), 50.1 (C-17), 17.2 (C-18), 17.0 (C-19), 86.2 (C-20), 24.0 (C-21), 34.1 (C-22), 26.3 (C-23), 84.2 (C-24), 71.2 (C-25), 26.1 (C-26), 28.4 (C-27), 23.6 (C-28), 30.4 (C-29), 16.8 (C-30), 101.4 (C-1′), 75.3 (C-2′), 78.4 (C-3′), 77.4 (C-4′), 72.8 (C-5′), 18.8 (C-6′), 102.5 (C-1″), 75.3 (C-2″), 78.7 (C-3″), 77.6 (C-4″), 73.1 (C-5″), 18.9 (C-6″).

Cyclocarioside J (**7**): white amorphous powder. ^1^H NMR (pyridine-*d*_5_, 600 MHz) *δ*_H_ 3.11 (1H, m, H-1a), 1.98 (1H, m, H-1b), 1.87 (1H, m, H-2a), 1.69 (1H, m, H-2b), 3.65 (1H, t, *J* = 3.7 Hz, H-3), 1.65 (1H, m, H-5), 1.48 (1H, m, H-6a), 1.49 (1H, m, H-6b), 4.33 (1H, dt, *J* = 5.3, 10.8 Hz, H-11), 1.04 (3H, s, H-18), 1.42 (3H, s, H-19), 1.10 (3H, s, H-21), 1.55 (3H, s, H-26), 1.39 (3H, s, H-27), 0.98 (3H, s, H-28), 1.30 (3H, s, H-29), 0.67 (3H, s, H-30), 4.75 (1H, d, *J* = 6.8 Hz, H-1′), 4.42 (1H, t, *J* = 7.6 Hz, H-2′), 4.21 (1H, dd, *J* = 2.1, 8.5 Hz, H-3′), 4.25 (1H, dd, *J* = 5.1, 9.5 Hz, H-4′), 4.35 (1H, m, H-5a′), 3.78 (1H, dd, *J* = 11.0, 13.4 Hz, H-5b′), 5.01 (1H, d, *J* = 7.1 Hz, H-1″), 4.12 (1H, m, H-2″), 4.21 (1H, dd, *J* = 4.8, 10.9 Hz, H-3″), 1.65 (3H, d, *J* = 5.7 Hz, H-6″). ^13^C NMR (pyridine-*d*_5_, 150 MHz) *δ*_C_ 36.2 (C-1), 21.5 (C-2), 81.5 (C-3), 38.1 (C-4), 51.2 (C-5), 18.4 (C-6), 36.5 (C-7), 50.1 (C-8), 54.4 (C-9), 40.2 (C-10), 76.5 (C-11), 34.3 (C-12), 41.2 (C-13), 41.2 (C-14), 31.2 (C-15), 26.9 (C-16), 49.9 (C-17), 17.0 (C-18), 16.9 (C-19), 85.8 (C-20), 24.4 (C-21), 34.5 (C-22), 25.8 (C-23), 84.0 (C-24), 71.1 (C-25), 26.1 (C-26), 28.6 (C-27), 23.3 (C-28), 30.4 (C-29), 16.6 (C-30), 103.4 (C-1′), 72.3 (C-2′), 75.4 (C-3′), 70.0 (C-4′), 67.8 (C-5′), 102.2 (C-1″), 76.0 (C-2″), 78.7 (C-3″), 77.1 (C-4″), 73.0 (C-5″), 19.0 (C-6″).

Cyclocarioside Z_18_ (**8**): white amorphous powder. ^1^H NMR (pyridine-*d*_5_, 600 MHz) *δ*_H_ 3.05 (1H, m, H-1a), 1.84 (1H, m, H-1b), 1.77 (1H, m, H-2a), 1.67 (1H, m, H-2b), 3.55 (1H, br s, H-3), 1.56 (1H, m, H-5), 1.58 (1H, m, H-6a), 1.44 (1H, m, H-6b), 4.43 (1H, dt, *J* = 5.0, 10.5 Hz, H-11), 1.05 (3H, s, H-18), 1.32 (3H, s, H-19), 1.45 (3H, s, H-21), 1.35 (3H, s, H-26), 1.35 (3H, s, H-27), 0.98 (3H, s, H-28), 1.26 (3H, s, H-29), 0.80 (3H, s, H-30), 5.51 (1H, br s, H-1′), 4.45 (1H, m, H-2′), 4.63 (1H, m, H-3′), 4.75 (1H, m, H-4′), 4.61 (1H, m, H-5a′), 4.28 (1H, m, H-5b′), 5.11 (1H, d, *J* = 7.5 Hz, H-1″), 4.02 (1H, d, *J* = 8.5 Hz, H-2″), 4.26 (1H, m, H-3″), 4.03 (1H, dd, *J* = 4.1, 8.4 Hz, H-4″), 4.17 (1H, t, *J* = 8.9 Hz, H-5″), 4.65 (H, d, *J* = 11.2 Hz, H-6a″), 4.35 (H, m, H-6b″). ^13^C NMR (pyridine-*d*_5_, 150 MHz) *δ*_C_ 36.2 (C-1), 21.6 (C-2), 79.5 (C-3), 38.1 (C-4), 51.2 (C-5), 18.6 (C-6), 36.5 (C-7), 50.1 (C-8), 54.4 (C-9), 40.7 (C-10), 77.5 (C-11), 34.3 (C-12), 41.4 (C-13), 41.6 (C-14), 31.7 (C-15), 26.8 (C-16), 49.6 (C-17), 17.2 (C-18), 16.9 (C-19), 86.8 (C-20), 24.5 (C-21), 34.5 (C-22), 26.8 (C-23), 84.5 (C-24), 71.1 (C-25), 26.1 (C-26), 28.3 (C-27), 23.3 (C-28), 30.4 (C-29), 16.9 (C-30), 106.4 (C-1′), 84.5 (C-2′), 80.0 (C-3′), 81.2 (C-4′), 65.8 (C-5′), 102.2 (C-1″), 76.0 (C-2″), 78.9 (C-3″), 78.1 (C-4″), 73.0 (C-5″), 19.6 (C-6″).

Cyclocarioside Z_10_ (**9**): white amorphous powder. ^1^H NMR (pyridine-*d*_5_, 400 MHz) *δ*_H_ 3.08 (1H, dd, *J* = 2.7, 9.2 Hz, H-1a), 2.12 (1H, m, H-1b), 2.11 (1H, m, H-2a), 1.79 (1H, m, H-2b), 3.63 (1H, m, H-3), 1.83 (1H, m, H-5), 1.53 (1H, m, H-6a), 1.45 (1H, m, H-6b), 4.46 (1H, dt, *J* = 5.7, 9.9 Hz, H-11), 1.12 (3H, s, H-18), 1.38 (3H, s, H-19), 1.44 (3H, s, H-21), 5.30 (1H, s, H-26a), 4.95 (1H, s, H-26b), 1.90 (3H, s, H-27), 0.98 (3H, s, H-28), 1.23 (3H, s, H-29), 0.85 (3H, s, H-30), 4.95 (1H, d, *J* = 7.5 Hz, H-1″), 3.97 (1H, t, *J* = 8.5 Hz, H-2″), 2.00 (3H, s, CH_3_COO), 4.96 (1H, br s, H-1′), 4.05 (1H, m, H-2′), 3.89 (1H, dd, *J* = 6.3, 7.5 Hz, H-3′), 4.75 (1H, m, H-4′), 4.21 (1H, dd, *J* = 3.3, 11.5 Hz, H-5a′), 4.28 (1H, dd, *J* = 6.3, 11.3 Hz, H-5b′), 4.40 (1H, d, *J* = 7.7 Hz, H-1″), 3.10 (1H, dd, *J* = 7.7, 9.0 Hz, H-2″), 3.35 (1H, dd, *J* = 8.7,9.0 Hz, H-3″), 2.98 (1H, dd, *J* = 2.7, 8.7 Hz, H-4″), 3.24 (1H, m, H-5″), 1.24 (3H, d, *J* = 5.8 Hz, H-6″). ^13^C NMR (pyridine-*d*_5_, 100 MHz) *δ*_C_ 37.1 (C-1), 26.3 (C-2), 79.5 (C-3), 38.2 (C-4), 51.0 (C-5), 18.7 (C-6), 36.6 (C-7), 41.6 (C-8), 54.7 (C-9), 40.3 (C-10), 34.2 (C-11), 76.9 (C-12), 41.6 (C-13), 50.0 (C-14), 31.7 (C-15), 27.7 (C-16), 50.1 (C-17), 17.1 (C-18), 16.8 (C-19), 86.6 (C-20), 24.4 (C-21), 34.1 (C-22), 26.3 (C-23), 84.2 (C-24), 71.5 (C-25), 26.5 (C-26), 26.4 (C-27), 29.6 (C-28), 22.4 (C-29), 16.9 (C-30), 106.4 (C-1′), 83.3 (C-2′), 79.4 (C-3′), 82.0 (C-4′), 65.8 (C-5′), 170.9 (CH_3_COO), 100.9 (C-1″), 75.1 (C-2″), 78.2 (C-3″), 77.1 (C-4″), 72.1 (C-5″), 18.3 (C-6″).

Cyclocarioside H (**10**): white amorphous powder. ^1^H NMR (pyridine-*d*_5_, 400 MHz) *δ*_H_ 2.58 (1H, dd, *J* = 3.7, 13.2 Hz, H-1a), 1.34 (1H, m, H-1b), 4.06 (1H, m, H-12), 1.10 (3H, s, H-18), 1.18 (3H, s, H-19), 1.14 (3H, s, H-21), 3.74 (1H, d, *J* = 7.5 Hz, H-24), 1.15 (3H, s, H-26), 1.19 (3H, s, H-27), 0.98 (3H, s, H-28), 0.90 (3H, s, H-29), 0.95 (3H, s, H-30), 4.95 (1H, d, *J* = 7.5 Hz, H-1″), 4.08 (1H, t, *J* = 8.5 Hz, H-2″), 4.08 (1H, d, *J* = 8.7 Hz, H-3″), 3.65 (1H, dd, *J* = 2.7, 8.7 Hz, H-4″), 3.60 (1H, d, *J* = 5.9 Hz, H-5″), 1.60 (3H, d, *J* = 5.8 Hz, H-6″). ^13^C NMR (pyridine-*d*_5_, 100 MHz) *δ*_C_ 36.1 (C-1), 26.4 (C-2), 75.5 (C-3), 38.9 (C-4), 50.9 (C-5), 19.0 (C-6), 36.0 (C-7), 50.6 (C-8), 54.7 (C-9), 40.3 (C-10), 77.2 (C-11), 35.0 (C-12), 41.6 (C-13), 42.0 (C-14), 31.7 (C-15), 27.7 (C-16), 50.1 (C-17), 17.5 (C-18), 17.1 (C-19), 74.6 (C-20), 27.4 (C-21), 38.1 (C-22), 31.3 (C-23), 76.2 (C-24), 150.1 (C-25), 110.5 (C-26), 18.9 (C-27), 23.6 (C-28), 30.4 (C-29), 16.9 (C-30), 101.9 (C-1″), 76.1 (C-2″), 78.7 (C-3″), 77.1 (C-4″), 73.1 (C-5″), 18.9 (C-6″).

2*α*,3*α*,23-trihydroxy-12,20(30)-dien-28-ursolic acid 28-*O*-*β*-D-glucopyranoside (**11**): ^1^H NMR (MeOD, 600 MHz) *δ*_H_ 5.28 (1H, t, *J* = 3.5 Hz, H-12), 4.71 (1H, br s, H-30a), 4.61 (1H, br s, H-30b), 3.78 (1H, m, H-23a), 3.70 (1H, dd, *J* = 11.6, 4.6 Hz, H-23b), 1.23 (s, 3H, H-26), 1.01 (3H, s, H-27), 0.99 (3H, s, H-25), 0.82 (3H, s, H-29), 0.80 (3H, s, H-24), 5.26 (1H, d, *J* = 8.4 Hz, H-1′), 3.85 (1H, m, H-6′a), 3.67 (1H, m, H-6′b). ^13^C NMR (MeOD, 150 MHz) *δ*_C_ 43.4 (C-1), 68.9 (C-2), 78.0 (C-3), 42.4 (C-4), 44.2 (C-5), 19.3 (C-6), 33.5 (C-7), 41.0 (C-8), 49.3 (C-9), 38.9 (C-10), 24.5 (C-11), 127.2 (C-12), 139.3 (C-13), 42.1 (C-14), 29.0 (C-15), 25.2 (C-16), 49.6 (C-17), 56.3 (C-18), 38.5 (C-19), 154.3 (C-20), 33.2 (C-21), 39.5 (C-22), 71.0 (C-23), 17.7 (C-24), 17.2 (C-25), 17.5 (C-26), 24.0 (C-27), 177.1 (C-28), 16.5 (C-29), 105.2 (C-30), 95.5 (C-1′), 74.1 (C-2′), 78.6 (C-3′), 71.3 (C-4′), 78.5 (C-5′), 62.0 (C-6′).

1-oxo-3*β*,23-dihydroxy-olean-12-en-28-oicacid-28-*O-β*-D-glucopyranosside (**12**): colorless amorphous power. ^1^H NMR (MeOD, 400 MHz) *δ*_H_ 5.15 (1H, t, *J* = 4.5 Hz, H-12), 3.72 (1H, m, H-3), 1.20 (3H, s, H-27), 0.93 (3H, s, H-30), 0.92 (3H, s, H-29), 0.85 (6H, s, H-24, 26), 5.33 (1H, d, *J* = 8.0 Hz, H-1′), 3.48–3.69 (6H, m, H-2′, H-3′, H-4′, H-5′, H-6′). ^13^C NMR (MeOD, 100 MHz) *δ*_C_ 215.0 (C-1), 44.6 (C-2), 73.4 (C-3), 44.1 (C-4), 47.4 (C-5), 18.3 (C-6), 33.2 (C-7), 40.7 (C-8), 40.3 (C-9), 53.1 (C-10),26.2 (C-11), 124.2 (C-12), 144.1 (C-13), 43.3 (C-14), 28.9 (C-15), 24.2 (C-16), 48.3 (C-17), 43.0 (C-18), 47.1 (C-19), 31.6 (C-20), 35.1 (C-21), 33.3 (C-22), 65.7 (C-23), 13.3 (C-24), 15.8 (C-25), 18.2 (C-26), 26.2 (C-27), 178.0 (C-28), 33.0 (C-29), 23.9 (C-30), 95.6 (C-1′), 78.3 (C-5′), 78.0 (C-3′), 73.7 (C-2′), 71.1 (C-4′), 62.5 (C-6’).

### 3.6. Hydrolyses of Novel Compounds to Determine the Linking Sugars

Each new isolate was respectively dissolved in 1 M hydrochloric acid (HCl) and refluxed for 2 h at 80 °C. The reaction solutions were then fractionated with ethyl acetate (EtOAc, 3 times). The remaining aqueous phase was concentrated under vacuum to remove water (H_2_O). Analytical HPLC was conducted on a Waters 2695 instrument (Waters Corporation, Milford, MA, USA) using Waters 2424 ELSD as a detector. A GH0525046C18AQ column (5 μm, 4.6 × 250 mm, Sil Green) was used, which used formic acid: water = 0.1: 100 (*v*/*v*) as mobile phase. The column temperature was 35 °C. The HPLC spectrum of the standards of L-arabinose, D-glucopyranose and D-quinovose are shown in the Appendix A. The existence of L-arabinose was determined by HPLC test with an authentic sugar in the aqueous phase of compound Z_1_ (**1**) (Appendix A), while D-glucopyranose and the L-arabinose were determined by HPLC test with two authentic sugars in the aqueous phase of compounds Z_2_ (**2**) and Z_3_ (**3**) (Appendix A). The presence of L-arabinose and D-quinovose was found by HPLC test with standard sugars in the water phase of compound Z_4_ (**4**) (Appendix A).

### 3.7. Assay for the Glucose Uptake Effects of the Fraction and Dammarane Saponins

#### 3.7.1. Cell Viability Assay

The cell viability assay was determined by the 3-(4,5-dimethyl-2-thiazolyl) -2,5-diphenyl-2H-tetrazolium bromide (MTT) (Sigma-Aldrich). The 3T3-L1 adipocytes were treated at a density of 3000 cells/well in 96-well plates in DMEM with 10% FBS. After 24 h of incubation, the cells were exposed to the test extract (CPBE, 25 μg/mL) and compounds (10 μM), which were dissolved in serum-free DMEM for 24 h. Each well was filled with twenty microliters of 2 mg/mL MTT solution and incubated for 4 h at 37 ℃ in the dark. The reduction product, a formazan, was dissolved with DMSO and the absorbance was measured at 550 nm with a microplate reader.

#### 3.7.2. Differentiation of 3T3-L1 Adipocytes

3T3-L1 cells were purchased from Cell Culture Center, Institute of Basic Medical Sciences, Institute of Basic Medicine, Chinese Academy of Medical Sciences (Batch number 1101MOU-PUMC 0001551). The cells were cultured in DMEM with 10% calf serum, 100 U/mL penicillin, and 100 mg/mL streptomycin (HyClone) in 5% CO_2_ at 37 ℃. The growth media was changed to DMEM with 10% fetal bovine serum (HyClone) containing 1 μM dexamethasone (Sigma-Aldrich, St. Louis, MO, USA), 520 μM 3-isobutyl-1-methylxanthine (Sigma-Aldrich), and 1 μg/mL insulin (Beijing, China, Roche, Germany) after two days. After 48 h, the cells were grown in DMEM with 10% FBS, 1 μg/mL insulin, 100 U/mL penicillin, and 100 mg/mL streptomycin for 2 days of incubation. Every two days, the medium was replenished with fresh DMEM supplemented with 10% FBS medium until adipogenesis was induced.

#### 3.7.3. Measurement of Glucose Uptake Assay

The level of glucose uptake was assessed using a fluorescent derivative of glucose, 2-NBDG (Beijing, China, Invitrogen, OR, USA). The fully differentiated 3T3-L1 adipocytes were seeded on 96-well plates with glucose-free medium containing 10% FBS and 1 μg/mL insulin. After 24 h of incubation, the cells were treated with test extract (CPBE, 25 μg/mL) as well as compounds (10 μM) and rosiglitazone (ROS, 10 μM, as a positive control) in the presence or absence of 50 μM 2-NBDG. The cells were rinsed with phosphate-buffered saline (PBS) after 1 h of incubation, while cell lysates were treated with 70 μL of 1% Triton X-100 in PBS and 0.1 M K3PO4 for 10 min. To quantify 2-NBDG fluorescence, the fluorescence signal was measured at an excitation wavelength of 450 nm and an emission wavelength of 535 nm using a SpectraMax M5 microplate reader. The 2-NBDG signal was determined using a fluorescence microscope (Olympus ix70, Tokyo, Japan) to detect glucose uptake.

## 4. Conclusions

To identify compounds for promoting glucose uptake, the components of *C. paliurus* leaves were separated, resulting in the identification of twelve triterpenoid saponins (**1**–**12**), including four previously undescribed dammarane triterpenoid saponins. The docking study demonstrated that the most active compound strongly bonded with PTP1B, hydrogen bonds and hydrophobic interactions, verifying the importance of the sugar unit. Bioassays demonstrated that the dammarane triterpenoid saponins **6**–**7** and **10** strongly enhanced insulin-stimulated glucose uptake in 3T3-L1 adipocytes in a dose-dependent manner. Collectively, *C. paliurus* leaves contain abundant dammarane triterpenoid saponins that affect glucose uptake in 3T3-L1 adipocytes; this discovery could be meaningful for the development of new treatments for insulin resistance and hyperglycemia.

## Figures and Tables

**Figure 1 molecules-28-03294-f001:**
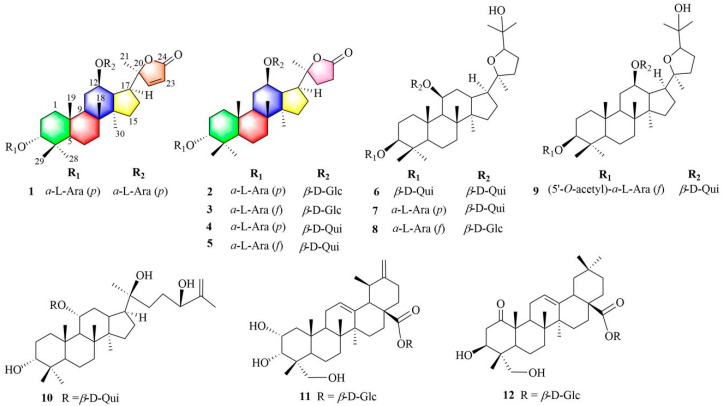
Structures of compounds **1**–**12** isolated from *C. paliurus*.

**Figure 2 molecules-28-03294-f002:**
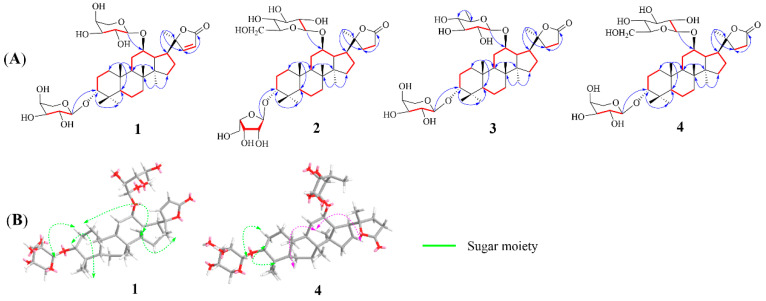
(**A**) Key HMBC (red arrows) and COSY (blue bold bonds) correlations of compounds **1**–**4**. (**B**) ROESY correlations of compounds **1** and **4**.

**Figure 3 molecules-28-03294-f003:**
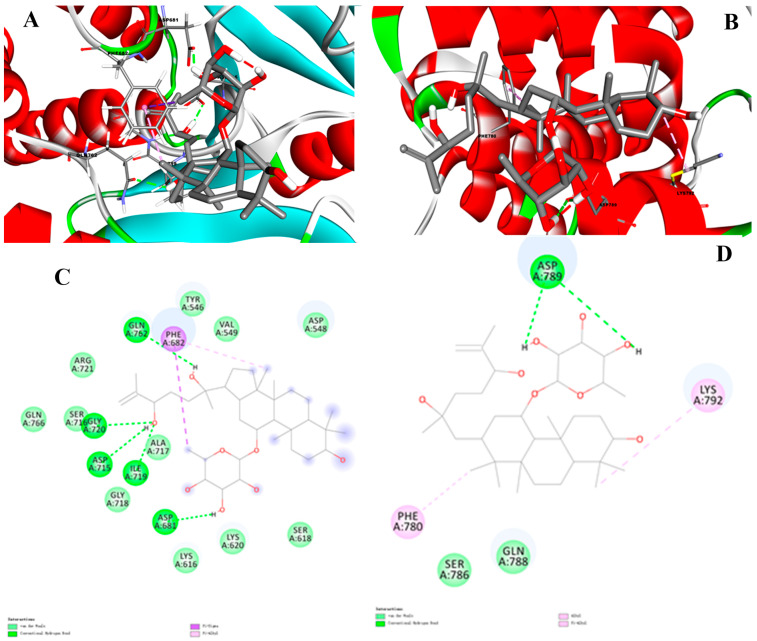
Catalytic site of three-dimensional (**A**) and two-dimensional (**C**) visualization of compound **10** against PTP1B; Allosteric sites of three-dimensional (**B**) and two-dimensional (**D**) visualization of compound **10** against PTP1B. Conventional hydrogen bond as green, Pi-sigma as purple, Pi-alkyl and alkyl interactions as light pink, and van der Waals as light green.

**Figure 4 molecules-28-03294-f004:**
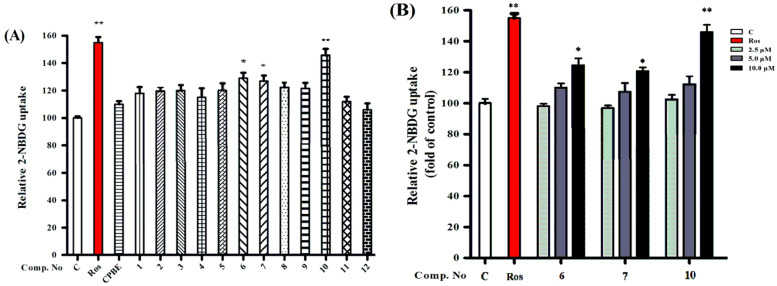
(**A**) The glucose uptake effects of compounds **1**–**12** on 3T3-L1 adipocytes using a fluorescent derivate of cells at 10 μM. (**B**) Compounds **6**–**7** and **10** increased glucose uptake in 3T3-L1 adipocytes. The differentiated 3T3-L1 adipocytes were treated with compounds **6**–**7** and **10** at various concentrations (2.5, 5.0, and 10 μM) for 1 h. These results are expressed as the mean ± SD (*n* = 3) of triplicate analysis; * *p* < 0.05 and ** *p* < 0.01.

**Table 1 molecules-28-03294-t001:** ^1^H and ^13^C NMR spectral data of compounds **1**–**2** (600 MHz, Pyridine-*d*_5_, *δ* in ppm).

Position	1		2	
	*δ*_H_, Mult (*J* in Hz)	*δ* _C_	*δ*_H_, Mult (*J* in Hz)	*δ* _C_
1	3.05, m; 2.00, m	36.1	3.13, m; 2.08, m	36.0
2	1.74, m; 1.68, m	22.2	1.87, m; 1.46, m	21.4
3	3.53, br s	81.6	3.60, br s	81.9
4		38.5		38.6
5	1.54, d, (11.2)	51.3	1.61, m	50.7
6	1.53, m; 1.40, m	18.7	1.58, m; 1.46, m	18.8
7	1.41, m; 1.06, d, (9.2)	36.8	1.48, m; 1.13, d, (11.7)	36.8
8		50.7		50.1
9	1.80, m	54.3	1.89, m	54.3
10		40.4		40.4
11	2.70, m; 1.46, m	34.1	2.65, m; 1.52, m	34.0
12	4.36, m	76.8	4.51, td, (10.6, 4.9)	76.4
13	1.80, m	42.5	1.73, m	41.6
14		41.9		41.0
15	1.32, m; 1.01, m	31.7	1.35, m; 0.97, m	31.6
16	1.64, m; 1.34, m	25.9	1.72, m; 1.43, m	26.0
17	1.53, m	47.8	1.49, m	50.6
18	0.90, s	17.5	0.98, s	17.5
19	1.29, s	17.4	1.38, s	17.4
20		92.6		89.9
21	1.36, s	23.9	1.28, s	24.9
22	7.64, d, (5.7)	161.2	2.00, m; 1.88, m	32.0
23	6.26, d, (5.7)	121.9	2.70, m; 2.59, m	29.1
24		173.4		176.8
28	0.91, s	23.5	0.96, s	23.8
29	1.20, s	30.4	1.26, s	30.0
30	0.55, s	16.9	0.62, s	16.8
1′	4.68, d, (7.2)	102.1	4.74, d, (6.5)	102.3
2′	4.38, dd, (9.2, 7.2)	72.9	4.45, t, (8.9)	72.1
3′	4.13, dd, (9.2, 4.2)	75.2	4.27, dd, (8.4, 3.5)	75.3
4′	4.19, dd, (8.2, 3.6)	70.3	4.42, m	69.7
5′	4.29, dd; (8.2, 3.6), 3.74, m	66.8	4.34, dd, (12.1, 3.7); 3.80, m	66.9
1′′	4.87, d, (6.5)	102.7	5.11, d, (7.7)	102.4
2′′	4.36, dd, (8.0, 7.2)	73.1	4.16, t, (8.2)	75.8
3′′	4.28, dd, (9.2, 4.1)	75.4	4.03, dd, (8.2, 6.5)	79.2
4′′	4.40, m	69.5	4.01, dd, (8.2, 6.5)	78.5
5′′	4.29, dd, (8.6, 3.2); 3.80, m	68.1	4.31, d, (8.9); 3.78, m	72.9
6′′			4.58, dd, (11.4, 2.8); 4.38, m	63.9

**Table 2 molecules-28-03294-t002:** The HPLC retention time of sugar moieties of compounds **1**–**4**.

Compounds	Retention Times (min)	Types of the Monosaccharides
**1**	6.844, 6.847	L-Ara, L-Ara
**2**	6.842, 6.562	L-Ara, D-Glc
**3**	6.849, 6.561	L-Ara, D-Glc
**4**	6.841, 7.419	L-Ara, D-Qui
D-Glc	6.567	
L-Ara	6.843	
D-Qui	7.416	

**Table 3 molecules-28-03294-t003:** ^1^H and ^13^C NMR spectral data of compounds **3**–**4** (600 MHz, pyridine-*d*_5_, *δ* in ppm).

Position	3		4	
	*δ*_H_, Mult. (*J* in Hz)	*δ* _C_	*δ*_H_, Mult. (*J* in Hz)	*δ* _C_
1	3.01, m; 2.01, m	36.6	3.10, m; 2.06, m	36.1
2	1.91, m; 1.74, m	22.3	1.89, m; 1.78, m	21.8
3	3.53, m	81.9	3.62, d, (5.1)	79.8
4		38.6		38.3
5	1.53, (d, 11.2)	51.4	1.60, m	51.3
6	1.60, m; 1.40, m	18.8	1.58, m; 1.46, m	18.6
7	1.39, m; 1.06, d, (9.2)	35.9	1.47, m; 1.44, m	36.5
8		50.7		50.6
9	1.80, m	55.4	1.89, m	54.2
10		40.4		40.3
11	1.98, m; 1.84, m	34.0	1.98, m; 1.84, m	34.1
12	4.16, m	76.0	4.51, m	76.5
13	1.61, m	42.3	1.73, m	42.0
14		41.0		41.6
15	1.32, m; 1.03, m	31.4	1.32, m; 1.03, m	31.5
16	1.64, m; 1.34, m	25.8	1.83, m; 1.72, m	25.8
17	1.53, m	50.0	1.81, m	49.9
18	0.90, s	17.1	1.03, s	17.2
19	1.34, s	17.4	1.38, s	17.1
20		89.7		89.9
21	1.47, s	21.9	1.25, s	23.3
22	1.98, m; 1.87, m	32.7	1.98, m; 1.84, m	32.6
23	2.67, m; 2.51, m	29.7	2.66, m; 2.50, m	29.7
24		176.6		177.3
28	0.97, s	23.6	1.00, s	23.2
29	1.28, s	30,4	1.28, s	30.3
30	0.69, s	16.7	0.61, s	16.7
1′	5.55, d, (6.5)	106.9	4.75, d, (6.5)	102.5
2′	4.89, t, (5.6)	84.6	4.44, m	72.9
3′	4.85, m	79.9	4.24, dd, (8.2, 3.5)	75.3
4′	4.76, dd, (6.0, 5.0)	86.0	4.40, t, (6.0)	69.5
5′	4.40, m; 4.29, d, (9.4)	63.4	4.33, dd, (12.1, 3.7); 3.79, m	66.9
1′′	5.12, d, (7.6)	102.1	5.00, d, (7.7)	101.9
2′′	4.03, t, (8.3)	75.0	4.07, t, (9.2)	75.7
3′′	4.31, t, (9.1)	78.9	4.18, t, (8.9)	79.0
4′′	4.05, m	77.4	3.70, t, (9.0)	78.3
5′′	4.18, dd, (9.1, 3.2); 3.78, m	73.4	4.41, m, 3.79; dd, (8.9, 2.7)	72.7
6′′	4.41, m; 4.30, m	63.8	1.66, d, (7.4)	19.1

## Data Availability

The data of the article can be obtained from the authors.

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
