# Peer review of "Triterpenoids from the Leaves of Cyclocarya paliurus and Their Glucose Uptake Activity in 3T3-L1 Adipocytes"

_molecules, 2023, doi:10.3390/molecules28083294_

Round 1
Reviewer 1 Report
The authors carried out a study entitled "Triterpenoids from the leaves of Cyclocarya paliurus and their glucose uptake activity in 3T3-L1 adipocytes"
In order for this manuscript to be recommended for publication, the authors must include more analysis, I have made a report where I show how the manuscript should improve.
If the authors are working with new or analogous molecules, the authors must insert a study on the physicochemical properties of these molecules, as this is not known, this study can be performed by molecular modeling.
Furthermore, I recommend that the authors propose a possible molecular interaction mechanism with a specific target.
Author Response
Dear reviewers,
On behalf of all the contributing authors, I would like to express our sincere appreciation of your letter and reviewers’ constructive comments concerning our entitled “Triterpenoids from the leaves of Cyclocarya paliurus and their glucose uptake activity in 3T3-L1 adipocytes” (manuscript No: ISSN 1420-3049). These comments are all valuable and help for improving our article. As you are concerned, there are several problems that need to be addressed. According to your nice suggestions, we have made extensive corrections to our previous draft, the changes to the manuscript are given in the red next.
Respectfully yours,
Jun Li
Our responses to the editor's and reviewers' comments for molecules-2292292 appear below. All the revised sections in the manuscript are highlighted in red.
Point 1: If the authors are working with new or analogous molecules, the authors must insert a study on the physicochemical properties of these molecules, as this is not known, this study can be performed by molecular modeling.
Furthermore, I recommend that the authors propose a possible molecular interaction mechanism with a specific target.
Response 1: Thank you for your comments. We think this is an excellent suggestion. We have made Molecular docking simulations to elucidate the possible molecular interaction mechanisms with respect to the pharmacological activities of dammarane triterpenoids, including the change can be found in the revised manuscript.

Reviewer 2 Report
English grammar check is needed for overall manuscript. There are a lot of spacing errors as well.
Figure 1 quality is lacking. The image quality is either on low definition, or need to change the quality.
Figure 1, from compounds 1-9, the author needs to reassigh the structure. For example, C-17 you can just draw an proton that is hashed wedged bond and not make the bold bond from C-17 to C-20.
Line 85, author stated the ROESY correlations have shown that H-3, H-11, H-19 are beta-oriented. Need more information to support this. ROESY correlations only shows the relative orientation of the structure. Author can suggest the biosynthetic characteristic of dammarane triterpenoids that H-19 (CH3) can only be B-oriented with a reasonable explanation.
From Lines 116 to 123, the order and the flow of the sentence is awkward. something needs to change.
General Experimental Procedues, need information of the column for C18 semipreparative HPLC. the size, length, and brand
Line 198, C. paliurus needs to be italicized
Regarding the supplementary data for HPLC spectrum of sugar moieties, the author did not show the spectrum for the isolated sugar moieties.
Author Response
Dear reviewers,
On behalf of all the contributing authors, I would like to express our sincere appreciation of your letter and reviewers’ constructive comments concerning our entitled “Triterpenoids from the leaves of Cyclocarya paliurus and their glucose uptake activity in 3T3-L1 adipocytes” (manuscript No: ISSN 1420-3049). These comments are all valuable and help for improving our article. As you are concerned, there are several problems that need to be addressed. According to your nice suggestions, we have made extensive corrections to our previous draft, the changes to the manuscript are given in the red next.
Respectfully yours,
Jun Li
Our responses to the editor's and reviewers' comments for molecules-2292292 appear below. All the revised sections in the manuscript are highlighted in red.
Point 1: English grammar check is needed for overall manuscript. There are a lot of spacing errors as well.
Response 1: We feel really sorry for our careless mistake. Thank you for your reminder, we have checked the English grammar and removed the spacing errors in overall manuscript.
Point 2: Figure 1 quality is lacking. The image quality is either on low definition, or need to change the quality.
Response 2: Thanks for your comments. We feel really sorry for our careless mistake. Thank you for your reminder. Figure 1 have been improved and uploaded again in our resubmitted manuscript.
Point 3: Figure 1, from compounds 1-9, the author needs to reassigh the structure. For example, C-17 you can just draw a proton that is hashed wedged bond and not make the bold bond from C-17 to C-20.
Response 3: Thank you for your kind reminder. According to your suggestion, we have changed the configuration at C-17 of compounds 1-9 in Figure 1.
Point 4 : Line 85, author stated the ROESY correlations have shown that H-3, H-11, H-19 are beta-oriented. Need more information to support this. ROESY correlations only shows the relative orientation of the structure. Author can suggest the biosynthetic characteristic of dammarane triterpenoids that H-19 (CH3) can only be B-oriented with a reasonable explanation.
Response 4: Thanks for your kind advice. As suggested by the revierwer, according to the biosynthetic characteristic of dammarane triterpenoids, all CH3-19 at position 10 of dammarane triterpenoids can only be β-oriented [35].
Point 5:From Lines 116 to 123, the order and the flow of the sentence is awkward. Some things needs to change.
Response 5: Thanks for your careful checks. We are sorry for our mistakes. From Lines 116 to 123, we have modified the sentences to make the sentences order are correct.
Point 6:General Experimental Procedues, need information of the column for C18 semipreparative HPLC. the size, length, and brand.
Response 6: thanks for your careful checks. We have Supplemented semipreparative HPLC of the size, length, and brand. Agilent 1260 semipreparative HPLC (ODS-A, 8 μm, 250 ×10.0 mm, YMC).
Point 7:Line 198, C. paliurus needs to be italicized.
Response 7: We sincerely thank the reviewer for careful reading. As suggested by the revierwer, we have corrected the “C. paliurus” into “C. paliurus”.
Point 8:Regarding the supplementary data for HPLC spectrum of sugar moieties, the author did not show the spectrum for the isolated sugar moieties.
Response 8: We feel really sorry for our carelessness. As suggested by the reviewer, we have supplemented HPLC spectra of the isolated sugar moieties in the Supporting Material.
Round 2
Reviewer 1 Report
The authors performed the revisions, I recommend the manuscript for publication
Reviewer 2 Report
Revision suggested are corrected.